# Serum Allergen-Specific Immunoglobulin E in Cats with Inflammatory Bronchial Disease

**DOI:** 10.3390/ani13203226

**Published:** 2023-10-15

**Authors:** Lina Hörner-Schmid, Jelena Palić, Ralf S. Mueller, Bianka Schulz

**Affiliations:** 1LMU Small Animal Clinic, University of Munich, 80539 Munich, Germany; 2Vet Med Labor GmbH Division of IDEXX Laboratories, 70806 Kornwestheim, Germany

**Keywords:** airways, feline asthma, chronic bronchitis, immunology, ELISA, allergy

## Abstract

**Simple Summary:**

To date, the etiology of feline chronic inflammatory bronchial disease, including feline asthma, chronic bronchitis, and mixed inflammatory forms, has not extensively been investigated. By measuring allergen-specific immunoglobulin E from the serum of cats with different types of inflammatory bronchial disease and healthy cats, we aimed to obtain information on serum reactions to environmental allergens. No significant differences were found in the number of positive immunoglobulin E-reactions between affected and healthy cats. Significantly, more cats with mixed airway inflammation reacted to mites. Sensitization to environmental allergens can be detected in healthy and diseased cats. Therefore, positive reactions must always be interpreted in the context of clinical signs and environmental conditions.

**Abstract:**

The etiology of feline inflammatory bronchial disease is poorly understood. This study compares the degree of allergen-specific serum IgE responses between cats with feline asthma, chronic bronchitis, mixed inflammation, and clinically healthy cats (HCs). The retrospective case–control study used serum from eighteen cats with eosinophilic inflammation (EI), ten with neutrophilic inflammation (NI), six with mixed inflammation (MI), and fourteen HCs. Affected cats were categorized into groups based on bronchoalveolar lavage cytology. The measurement of IgE for 34 different allergens including fungal organisms, weeds, grasses, trees, mites, and insects was performed using an indirect ELISA. Positive reactions to allergens were detected in the serum of 17/18 cats with EI, 8/10 with NI, 6/6 with MI, and 11/14 HCs (*p* = 0.364). When overall positive reactions were compared between groups, cats with MI (*p* = <0.01) had significantly more positive reactions against mite allergens than HCs. Blood eosinophils inversely correlated with the absolute amount of allergen-specific serum IgE expressed in ELISA absorbance units (EAs) (*p* = 0.014). Sensitization against dust mites seems to be more prevalent in cats with MI. However, positive IgE reactions can be observed in healthy and diseased cats, and, therefore, need to be interpreted in the light of clinical findings and environmental conditions of individual patients.

## 1. Introduction

Feline chronic inflammatory bronchial disease (FBD) affects approximately 1% of the cat population [1,2,3,4,5]. Previously, a categorization of FBD into chronic bronchitis (CB), feline asthma (FA), and mixed inflammation (MI) has been proposed [3,5,6,7,8]. It is unknown if this classification represents separate disease entities or simply variations of the same disease. To date, FA, CB, and MI can only be differentiated by a cytological evaluation of bronchoalveolar lavage fluid (BALF). Based on cytological classification and the consideration of clinical, radiographic, and functional aspects, neutrophilic inflammation in BALF is suggestive of CB, whereas eosinophilic inflammation indicates FA [6,7]. However, studies were not able to detect clinical, hematological, or radiographic differences between the different inflammatory conditions [3,5].

FA is considered a type-II hypersensitivity reaction [1,6,8,9,10,11]. A T-helper 2 (TH2) immune response with the secretion of interleukin-4 (IL-4), IL-5, IL-6, and IL-13 in response to inhalation of allergens has been described [8,11,12]. In one study, mono- or polysensitization against aeroallergens has been detected in 78% of naturally affected cats with FA [13]. Studies demonstrated more positive immunoglobulin E (IgE) responses in cats with FBD compared to healthy ones [14,15]. Allergen-specific IgE can be measured using in vivo (e.g., intradermal tests (IDTs)) or in vitro tests (e.g., monoclonal antibody tests). IDTs are not always suitable due to the need for a large shaving area, sedation, risk of anaphylaxis, and difficulties in interpreting the response [14,16,17]. Allergen-specific serum IgE tests are, therefore, easier to perform and proved to be highly specific in an experimental model of FA [18].

CB has not been the subject of many investigations in cats [3,5,19]. In human medicine, cigarette smoke, exposure to pollutants, and previous airway infections have been identified as predisposing factors for CB [20,21,22,23,24]. Similar speculations exist in veterinary medicine [25,26]. The neutrophilic airway inflammation in CB is likely induced by a T-helper-17 (TH17) cell cytokine pattern with an increased expression of IL-17A, IL-22, and IL-23 [27,28]. In human medicine, increased IgE serum concentrations have been described in patients with CB [29,30]. In research cats, allergen sensitization induced not just an asthmatic phenotype but also a neutrophilic or mixed airway inflammation [7,31]. MI has rarely been described in cats [5,32], and non-eosinophilic asthma remains a poorly understood condition, even in human medicine [33].

Investigation of the etiologies of different types of airway inflammation might be important to develop subsequent treatment strategies. Allergen (-specific) immunotherapy (AIT) in cats with induced asthma improved clinical signs and decreased BALF eosinophils, as well as IL-4, and IL-5 [34,35,36,37]. In another study, AIT positively influenced clinical signs in naturally affected cats with FA [38].

The aim of this study was to evaluate allergen-specific serum IgE responses in cats with eosinophilic inflammation (EI), neutrophilic inflammation (NI), MI, and clinically healthy cats (HCs) and compare the results between groups. In addition, a potential relationship between the number of blood eosinophils and IgE reactions was investigated.

## 2. Materials and Methods

This study was designed as a retrospective clinical case–control study. Stored serum from cats presented to the LMU Small Animal Clinic, University of Munich, from January 2017 to July 2020, was used for the study. This study was approved by the Ethics Committee of the Centre for Clinical Veterinary Medicine of the LMU University of Munich (No. 260-08-03-2021).

### 2.1. Study Population and Diagnostic Tests

Stored serum was used from 34 client-owned cats with FBD that had prospectively been included in previous studies and that had been categorized into the following groups: 18 cats with EI, 10 cats with NI, and 6 cats with MI. Inclusion criteria were a history of typical clinical signs of FBD (cough, wheezing, and/or dyspnea), radiographic findings consistent with a bronchial or bronchointerstitial pattern, and complete BALF results including bacteriologic examination (aerobe culture), Mycoplasma spp. PCR, and cytological examination. Only serum from culture- and Mycoplasma-negative cats was included. Bronchoalveolar lavage (BAL) was performed in all cats as part of the diagnostic work-up as a “blind” procedure, as previously described [39]. Based on the BALF cytology, differential count cats were categorized into three different groups, as previously described [5]. Cats were diagnosed with EI if they had >20% eosinophils and <14% neutrophils or if >50% eosinophils were present in BALF-cytology. NI was defined as >14% neutrophils and <20% eosinophils, and MI was defined as 20–50% eosinophils and >14% neutrophils. In addition, the results of complete blood counts were available from all cats with FBD. Baermann fecal analysis from a 3-day fecal sample was performed in 8 cats with outdoor access. None of the cats had been in other countries; therefore, testing for dirofilaria immitis was not performed, as this parasite is not endemic in Germany. The leftover serum from 14 HCs was included for the control. In these cats, serum had been collected for other diagnostic reasons (e.g., general check-up, vaccine titer check, etc.). Cats were assigned to the control group if they showed no abnormalities on clinical examination and had no history of respiratory signs, heart disease, skin disease, or gastrointestinal disease.

### 2.2. Allergen Testing

Serum samples had been frozen at −80 °C until analysis. For allergen panels, an ELISA with monoclonal antibodies derived from recombinant IgE was used (ArtuVet, Lelystad, The Netherlands). All sera were tested for specific IgE against cross-reactive carbohydrate determinants (CCDs) by indirect ELISA (CCD screening) in the first step. Briefly, sera were diluted 1/6 in a dilution buffer containing 1% bovine serum albumin (BSA) and 0.1% Tween^®^ 20 detergent. Once diluted, sera were added to plates coated with the CCDs and blocked with a blocking buffer (1% BSA and 0.1% Tween 20) for 30 min before sera were added. After overnight incubation at 4 °C, plates were washed four times, and olygo.3mAB a (mixture of 3 monoclonal antibodies produced against a recombinant dog IgE) was added to the wells. Plates were kept for 2 h at 4 °C and, after washing six times with washing buffer, a para-nitrophenylphosphat (pNPP) substrate was added to the wells. After 30 min at room temperature, the reaction was stopped by adding natriumhydroxid 1N. Finally, absorbance was measured at 405 nm.

Sera that tested negative against CCDs were analyzed, as described above using plates coated with different allergen extracts. In cases in which specific IgE against CCDs were detected, sera were blocked with a semi-synthetic CCD blocker at 20 µg/mL before dilution. After this blocking step, the samples were analyzed, as already described.

The amount of allergen-specific serum IgE was expressed in ELISA absorbance units (EAs). Results < 150 EA were considered negative. Values of 150–200 EA were defined as intermediate, and values > 200 EA were defined as positive reactions. Thirty-four different allergens were tested. Allergens belonged to the subgroups of mites and the pollen of crops, weeds, trees, grasses, fungi, and others (Table 1). Indoor allergens included groups of mites, fungi, and others. Environmental allergens consist of pollen from crops, weeds, trees, and grasses.

### 2.3. Statistical Analysis

Data were tested for normal distribution by the D’Agostino and Pearson omnibus tests. Normally distributed clinical data and indoor and outdoor access were compared between cats with FBD and HCs with a *t*-test. Non-normally distributed data were compared with the Mann–Whitney U-test. The following aspects were statistically compared by one-way ANOVA on the ranks between all four groups of cats. 1. The number of positive serum IgE reactions; 2. the number of positive and intermediate reactions; 3. The absolute values of the allergen-specific serum IgE in EA; and 4. blood eosinophils per µL. In addition, aspects 1 to 3 were compared between two cat groups with a *t*-test, if normally distributed. The correlation of blood eosinophils in cells/µL and BALF eosinophils in % and serum IgE in EA was tested for significance with a Spearman rho test. The presence of CCD antibodies and negative results were compared between groups using chi-square analysis. Groups of allergens were compared for positive reactions between groups of cats using Fisher’s exact test. To compare the number of positive serum IgE reactions, the number of positive and intermediate reactions, and the absolute values of reactions in EA and allergen groups, an individual significance level of *p* < 0.007 was applied to the data by the Bonferroni method for multiple comparisons. For the statistical comparison of CCD blocking between the different types of inflammation, a significance level of *p* < 0.01 was calculated, according to the Bonferroni method. A *p*-value of <0.05 was considered significant for all tests. All analyses were performed using GraphPad Prism 6.0 (GraphPad Software Inc.; San Diego, CA, USA).

## 3. Results

### 3.1. Study Cats

The study included 34 cats with FBD and 14 HCs. The mean age (with standard deviation (SD)) did not differ significantly between cats with NI (9.1 ± 3.8, *p* = 0.55), MI (8.2 ± 5.6, *p* = 0.26), and HCs (11.1 ± 4.6). There was a significant difference in age between HCs and cats with EI (5.4 ± 3.2, *p* < 0.01). Breeds of cats with FBD were European Shorthair (18), Abyssinian (2), Siamese (2), mixed breed cats (3), Maine Coon (1), British Shorthair (2), Ragdoll (2), Turkish Van (1), Somali (1), Siberian Forest Cat (1), and Bengal (1). The 14 HCs included European Shorthair cats (12), a Siberian Forest cat (1), and a Siamese mix (1). Two cats (one with EI and one with NI) had been pretreated with inhaled corticosteroids (fluticasone) at the time of work-up.

### 3.2. Allergen Reaction

A total of 17/18 cats with EI, 8/10 cats with NI, and all cats with MI had at least one positive result with the allergens tested. A total of 11/14 HCs had at least one positive IgE response. The number of negative responses between groups did not differ significantly (*p* = 0.364). One cat with EI and one with NI showed monosensitization to one single allergen. In the HC group, two cats had monosensitization, while in the MI group, all cats had polysensitization.

IgE results were compared between groups using relative and absolute values. The absolute reactions were defined as the dimension of IgE reactions in EA (e.g., 522 EA for Phleum pratense). The relative values were defined as the number of positive IgE reactions (e.g., one positive reaction for Phleum pratense).

Cats with FA, CB, MI, and HCs did not differ significantly in the total number of positive IgE reactions (>200 EA) (*p* = 0.523). A detailed comparison of the positive reactions between groups of cats is shown in Table 2. There were no significant differences when single groups were compared for the number of positive IgE reactions.

In addition, it was evaluated if there was a difference between groups regarding responses to allergens when responses classified as intermediate also accounted as positive. Adding intermediate (150–200 EA) to the positive reactions (>200 EA) did not reveal a statistically significant difference between groups (EI 99/567, NI 50/325, MI 36/182, HCs 49/440; *p* = 0.537). The absolute extent of serum IgE responses measured in EA values also showed no significant difference between all four groups (*p* = 0.438). Absolute serum IgE responses in the EA values for the different cat groups are displayed in Table 3.

When negative IgE reactions of all cats with FBD were compared to HCs, no significant difference was found (*p* = 0.363).

### 3.3. Different Allergen Groups

There were 7/18 cats with outdoor access with EI, 3/10 with NI, and 1/5 with MI. When positive reactions to indoor allergens were compared between indoor only and indoor/outdoor cats with FBD, the difference was not statistically significant (*p* = 0.7). There was also no significant difference between groups with respect to outdoor allergens (*p* = 0.48).

While all groups of cats with FBD had the most positive reactions in the mite allergen group, HCs reacted most commonly to grasses. Reactions to mite allergens were significantly more common in cats with MI (14/30 positive reactions, *p* 0.006) compared to HCs (13/70). Data for allergen reactions in all four groups of cats are shown in Table 4.

A reaction against *Dermatophagoides farinae* was seen most frequently in 23/34 cats with FBD and 30/48 cats of the total population. All positive serum IgE reactions to single allergens in all groups of cats are shown in Table 5.

Positive reactions for CCDs were seen in 6/18 cats with EI, 2/10 cats with NI, 4/6 cats with MI, and 2/14 HCs (*p* = 0.304).

In 14 of 34 cats with FBD, clinical signs had been present for less than one year; therefore, a seasonal influence on clinical signs could not be assessed in these cats. Six cats (three EI, three NI) were described by owners as seasonal, four (one EI, one NI, two MI) were considered non-seasonal, and no data regarding seasonality were available for ten cats. In four of thirty-four cats (three EI, one MI), food-responsive chronic enteropathy was reported. A history of atopic dermatitis was present in three cats (two NI, one MI). Twenty-two cats lived in urban environments and eight lived in rural environments. The positive responses in these cats did not differ significantly when compared (*p* = 0.303).

### 3.4. Blood and BALF Eosinophils

Mean blood eosinophils were not significantly different when the three groups of cats with FBD were compared (*p* = 0.146). The absolute EA values in the groups of cats with FBD were inversely correlated with the number of blood eosinophils (*p* = 0.014, R = −0.44). Absolute EA values did not correlate with the percentage of eosinophils in BALF cytology (*p* = 0.112).

## 4. Discussion

In this study, the serum of cats with different types of airway inflammation was evaluated for allergen-specific IgE reactions for the first time to the authors’ knowledge. It could be demonstrated that cats with MI were more likely to have an IgE response against mite allergens compared to HCs; however, this group did not have more single positive IgE reactions compared to the other groups. In contrast to these results, in a pilot study, including 10 cats with non-classified lower airway disease and 10 healthy control cats, a significant difference was detected for positive IgE reactions between groups [14]. Similarly, another study with comparable numbers of cats showed significantly more positive IgE reactions between cats with EI and HCs [15]. However, that study included animals with more than 7% neutrophils in the MI group. In the present study, cats with BALF neutrophils between 7 and 14% were still included in the EI group. Therefore, it seems possible that similar results would have been obtained in the present study if different cut-off values for BALF cytology had been used. Another reason for the varying results in different studies could be different measurement methods. In previous studies, the FcεR1-ELISA had been used for the detection of allergens [13,14,15], while in the present study, an ELISA with monoclonal antibodies derived from recombinant IgE was used. In all studies, including the present one, no significant difference could be detected for the number of cats with positive serum IgE responses [14,15] or the negative responses among cats with FBD and HCs [15].

In the present study, the most prevalent positive reactions were seen against *Dermatophagoides farinae* and *Dermatophagoides pteronyssinus*. This is a finding that was previously reported in studies investigating sensitization to allergens in cats with EI and cats with non-classified FBD [13,14,15]. Interestingly, in the study that performed the IDT and IgE tests in cats with FBD, significantly more positive reactions to *Dermatophagoides farinae* were detected in the IDT than in the IgE test [14]. In the present study, a response to mites was significantly more common in cats with MI than in HCs. High concentrations of dust mites can be found in the environment of indoor cats, which leads to a high risk of sensitization [40]. *Dermatophagoides* spp. also appear to induce hypersensitivity reactions in horses, humans, and dogs [41,42,43] and are the most frequently represented allergens in feline atopic dermatitis [44]. Although it could be assumed that cats with outdoor access might be sensitized less commonly to indoor allergens, this could not be confirmed by our results. No difference regarding reactions to indoor and outdoor allergens could be detected in cats with different housing types. This result is in agreement with observations of a previous study in cats with EI [13]. Another allergen commonly identified in the present study was *Tyrophagus putrescentiae*. This storage mite induced reactions in 29% of cats with FBD. Interestingly, a previous study reported that 2/20 cats with EI improved with diet change from dry to wet food and became clinically asymptomatic without additional treatment [38]. It is unknown whether the cats in the present study were fed dry or wet food. It would be interesting to further investigate the influence of food and storage mites on airway inflammation in the future.

Cats with FA were younger than cats with CB in several studies [3,5]. This was not the case in the present study; however, cats with FA were significantly younger than the control group. This study was the first to compare IgE responses between cats with different types of airway inflammation and HCs. In humans, IgE responses have been shown to decrease with age [45]. If this finding would apply to cats, a difference between cats with EI and HCs would be expected.

Interestingly, some cats with EI additionally suffered from gastrointestinal or skin diseases compatible with an allergic etiology. In another study on cats with lower airway disease, patients with evidence of allergic skin disease had been excluded from participation [14]. It is unknown if cats commonly had been excluded from other studies because of concomitant dermatologic or gastrointestinal disease and how an exclusion might have influenced the results of allergy testing. In the present study, these comorbidities were not considered a criterion for exclusion, as they support the theory of a possible allergic origin.

In previous studies, an allergen-specific IgE response had only been evaluated in cats with eosinophilic airway inflammation alone [13] or in comparison to a healthy control group [14,15]. In the present study, additional analyses in subgroups with NI and MI were performed. There was no significant difference between the groups regarding the total number of positive IgE responses. This observation raises the question of whether allergies are commonly involved in the etiology of feline inflammatory lower airway disease. In addition, it remains questionable if different forms of lower airway inflammation can really be considered different disease complexes or if they appear to be different manifestations of the same underlying disease since no clinical or radiographic differences could be detected between cats with EI, NI, and MI, and even BALF cytology can vary significantly among samples taken from different lung lobes [3,5,32].

When comparing the results between diseased and healthy individuals, it must be kept in mind that positive serum IgE responses reflect reactions to exposures that may not be clinically relevant. In human medicine, it has been postulated that atopy is overestimated as a factor in the development of asthma [46,47]. An asthmatic condition is thought to have a multifactorial origin that perpetuates an inflammatory process in the lower airways. Genetics, epigenetics, previous viral infections, microbiome composition, and other factors are thought to play a role in the etiology of asthma [23,30,48,49,50,51,52,53].

In human medicine, in vivo allergy diagnostics, such as the IDT, are considered the gold standard. In human and feline medicine, the IDT has been described as a highly sensitive method for the identification of sensitization to allergens [14,16]. However, serum IgE testing is less time-consuming than IDT, easier to perform, and does not require anesthesia in feline patients. To date, it still remains unknown whether results of the serum IgE test and the IDT correlate well in naturally diseased cats with FBD or if both test forms need to be performed to receive a complete picture of the sensitization pattern in individual patients. Nevertheless, the results of the IDT and IgE test must always be interpreted considering clinical signs and environmental factors [14,25,54].

This study is the first investigation that included the elimination of CCDs before the evaluation of IgE responses. CCDs are epitope structures of plant and insect glycoproteins. The formation of a CCD-specific IgE does not lead to cross-linking with subsequent mast cell degranulation due to the monovalent structure of CCDs. Anti-CCD IgE antibodies, therefore, have little clinical significance but may confound serological tests [55,56]. After blocking anti-CCD IgE, better agreement between the results of the IDT and serum IgE tests was observed in atopic dogs tested for grass and weed allergens [57]. In a previous study, the influence of CCD on reactions to trees, grasses, and weeds was found in 13% of cats with clinically suspected allergies [58]. In cats with positive reactions against grass allergens, a reevaluation of results after blocking CCDs has been recommended. In the present study, 35% of the cats tested positive for CCDs and were blocked. Therefore, previous studies without CCD blockage might have overestimated positive IgE reactions in comparison to the present study.

The association between positive IgE reactions and the percentage of eosinophils in BALF has been investigated in the present study and previous investigations without detecting a positive correlation [13,15]. In the present study, an inverse correlation was found between the level of blood eosinophils and the absolute EA levels in cats with positive IgE reactions. An explanation for this finding could be that elevated IgE levels might facilitate the degranulation of mast cells and thus stimulate the outflow of eosinophils from the peripheral blood into the airways. This could explain lower blood eosinophil levels in patients with higher EA levels.

This study has some limitations due to its retrospective nature and relatively small numbers of cats included in the different groups, which might have affected statistical power. Another point of criticism is the fact that recombinant anti-canine IgE was used for the IgE measurement. This condition is apparent because the folding and thus the function of the recombinant proteins for cats may differ from native proteins. Previous studies also used anti-canine IgE, as feline IgE has not been available [15]. Other studies use human FcEpsilon receptors to detect IgE [13,14]. In the future, it could be investigated whether IgE measurements with recombinant feline IgE achieve different values. In addition, in future prospective studies, it would be interesting to perform a standardized classification of the clinical signs to correlate a clinical score with serum IgE reactions. Furthermore, it could be interesting to investigate environmental factors and their influence on allergen sensitization in affected cats.

## 5. Conclusions

In this study, when investigating serum IgE responses against environmental allergens in cats with different types of inflammatory bronchial disease and healthy cats, no difference in the number of positive reactions could be found between all groups of cats. However, reactions against house dust mite aeroallergens were more prevalent in cats with MI. The measurement of allergen-specific serum IgE reactions may serve as a future tool to facilitate allergen avoidance and ASIT. Therefore, it should be considered as a general work-up of the airway.

## Figures and Tables

**Table 1 animals-13-03226-t001:** Allergens (common and Latin names) tested on the serum of cats with feline bronchial disease and healthy cats.

Mites	Pollen of Crops	Weeds	Trees	Grasses	Fungi	Other
Mould mite (*Tyrophagus putrescentiae*)	Rape (*Brassica napus*)	Nettle mix (*Urtica* spp.)	Silver birch (*Betula verrucosa*)	Orchard grass (*Dactylis glomerata*)	*Alternaria alternata*	Flea (*Ctenocephalides* spp.)
House dust mite (*Dermatophagoides farinae*)		Lambs quarter (*Chenopodium album*)	Olive (*Olea europaea*)	Timothy (*Phleum pratens*)	*Aspergillus fumigatus*	
Hay mite (*Lepidoglyphus destructor*)		Common ragweed (*Ambrosia elatior*)	London plan (*Platanus hispanica*)	Rye grass (*Lolium perenne*)	*Cladosporium herbarum*	
House dust mite (*Dermatophagoides pteronyssinus*)		Sheep sorrel (*Rumex acetosella*)	Osier (*Salix viminalis*)	Bermuda grass (*Cynodon dactylon*)	*Malassezia*	
Grain mite (*Acarus siro*)		Common mugwort (*Artemisia vulgaris*)	Pine (*Pinus sylvestris*)	Oat (*Avena sativa*)		
		Dandelion (*Taraxacum officinale*)	European privet (*Ligustrum vulgare*)	Rye (*Secale cereale*)		
		Englisch plantain (*Plantago lanceolata*)	Cypress (*Cupressus sempervirens*)	Blue grass, Kentucky (*Poa pratensis*)		
			English oak (*Quercus robur*)			
			American elm (*Ulmus Americana*)			

**Table 2 animals-13-03226-t002:** Comparison of positive immunoglobulin E-reactions between different cat groups with an indication of *p*-values.

	EI ^1^ 83/567	NI ^2^ 40/325	MI ^3^ 28/182	HC ^4^ 46/440
EI ^1^ 83/567		0.887	1.0	0.341
NI ^2^ 40/325	0.887		0.662	0.485
MI ^3^ 28/440	1.0	0.662		0.298
HCs ^4^ 46/440	0.341	0.485	0.298	

^1^ EI, eosinophilic inflammation ^2^ NI, neutrophilic inflammation ^3^ MI, mixed inflammation ^4^ HCs, healthy cats.

**Table 3 animals-13-03226-t003:** Absolute values of the immunoglobulin E-response in mean ELISA absorbance units compared between cat groups with indication of the *p*-values.

	EI ^1^ 2727.17	NI ^2^ 1410.31	MI ^3^ 1027.09	HC ^4^ 1845.01
EI ^1^ 2727.17		0.859	0.264	0.431
NI ^2^ 1410.31	0.859		0.264	0.62
MI ^3^ 1027.09	0.264	0.264		0.266
HCs ^4^ 1845.01	0.431	0.62	0.266	

^1^ EI, eosinophilic inflammation ^2^ NI, neutrophilic inflammation ^3^ MI, mixed inflammation ^4^ HCs, healthy cats.

**Table 4 animals-13-03226-t004:** Positive allergen-specific serum immunoglobulin E-results (>200 EA) for different allergen and patient groups and a comparison between groups (bold numbers indicating significance).

	EI ^1^ (*n* = 18)	NI ^2^ (*n* = 10)	MI ^3^ (*n* = 6)	HC ^4^ (*n* = 14)	*p*-Values						
					All groups	HCs vs. EI	HCs vs. NI	HCs vs. MI	EI vs. NI	EI vs. MI	NI vs. MI
Mites	30	15	14	13	0.032	0.048	0.189	**0.006**	0.710	0.198	0.155
Pollen of crops	1	0	1	0	0.324	1.0		0.3	1.0	0.446	0.375
Weeds	7	4	4	5	0.771	1.0	1.0	0.452	1.0	0.470	0.470
Trees	31	16	6	12	0.098	0.030	0.1	0.788	0.867	0.214	0.344
Grasses	26	16	5	22	0.499	0.746	1.0	0.168	0.720	0.255	0.212
Fungi	5	3	5	2	0.062	1.0	1.0	0.095	1.0	0.115	0.119
Other	1	0	1	1	0.634	1.0		0.263	1.0	0.447	0.357

^1^ EI, eosinophilic inflammation ^2^ NI, neutrophilic inflammation ^3^ MI, mixed inflammation ^4^ HCs, healthy cats.

**Table 5 animals-13-03226-t005:** Positive serum immunoglobulin E-reactions for single allergens in all patient groups.

	All cats with FBD (*n* = 34)	EI ^1^ (*n* = 18)	NI ^2^ (*n* = 10)	MI ^3^ (*n* = 6)	HCs ^4^ (*n* = 14)
House dust mite (*Dermatophagoides farinae*)	23	11	6	6	7
House dust mite (*Dermatophagoides pteronyssinus*)	13	7	4	2	4
European privet (*Ligustrum vulgare*)	13	9	3	1	2
Mould mite (*Tyrophagus putrescentiae*)	10	6	2	2	1
Oat (*Avena sativa*)	8	4	3	1	4
Osier (*Salix viminalis*)	8	1	2	0	2
Bermuda grass (*Cynodon dactylon*)	7	5	1	1	5
English oak (*Quercus robur*)	6	2	2	1	1
Timothy (*Phleum pratense*)	5	2	2	1	3
Cypress (*Cupressus sempervirens*)	5	2	3	0	5
*Alternaria alternata*	5	2	2	1	1
*Aspergillus fumigatus*	5	2	1	2	1
Orchard grass (*Dactylis glomerata*)	4	3	0	1	3
Rye grass (*Lolium perenne*)	4	2	1	1	0
Nettle mix (*Urtica* spp.)	4	2	1	1	0
Dandelion (*Taraxacum officinale*)	4	2	1	1	0
Rye (*Secale cereale*)	3	1	2	0	0
Blue grass, Kentucky (*Poa pratensis*)	3	2	1	0	2
Grain mite (*Acarus siro*)	3	1	0	2	0
Silver birch (*Betula verrucosa*)	3	3	1	0	0
Lambs quarter (*Chenopodium album*)	3	1	1	1	1
Hay mite (*Lepidoglyphus destructor*)	2	2	0	0	0
Rape (*Brassica napus*)	2	1	0	1	0
Englisch plantain (*Plantago lanceolata*)	2	1	1	0	1
Flea (*Ctenocephalides* sp.)	2	1	0	1	0
Pine (*Pinus sylvestris*)	1	1	0	0	1
American elm (*Ulmus Americana*)	1	1	0	0	0
Common ragweed (*Ambrosia elatior*)	1	1	0	1	0
*Cladosporium herbarum*	1	1	0	0	0
Olive (*Olea europaea*)	0	0	0	0	0
London plane (*Platanus hispanica*)	0	0	0	0	0
Sheep sorrel (*Rumex acetosella*)	0	0	0	0	1
Common mugwort (*Artemisia vulgaris*)	0	0	0	0	1
*Malassezia*	0	0	0	0	1

^1^ EI, eosinophilic inflammation ^2^ NI, neutrophilic inflammation ^3^ MI, mixed inflammation ^4^ HCs, healthy cats.

## Data Availability

The data presented in this study are available upon request from the corresponding author.

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
