# Peer review of "Serum Allergen-Specific Immunoglobulin E in Cats with Inflammatory Bronchial Disease"

_animals, 2023, doi:10.3390/ani13203226_

Round 1

Reviewer 1 Report

I appreciate the authors doing this study. I think they have pointed out many of the limitation, I would interested in the authors would recommend the  addition of this test for the general work-up of the airway cat. 

Author Response

We thank the reviewer for the valuable comments. According to your feedback we inserted the following sentence:

Line 353-354: “Therefore, it should be considered as a general work-up of the airway.”

Reviewer 2 Report

This study described that inflammatory bronchial disease in cats could be association with allergen specific IgE according to the types. The manuscript was well written. Only minor corrections are needed to be published.

1. Line 78: the full name of LMU should be identified. 

2. Did the authors repeatedly address 'cross-reactive carbohydrate determinants (CCD)' in a manner different from conventional ELISAs? This should be briefly described in the Methods section.

None.

Author Response

We thank the reviewer for the valuable comments.

  1. Thank you very much for this correction. We have adjusted the term.

Line 77-79: Stored serum from cats presented to the Ludwig-Maximilians-Universität (LMU) Small Animal Clinic, University of Munich, from January 2017 to July 2020 was used for the study.

  1. No, this was a completely normal blocking procedure.

Reviewer 3 Report

Authors describe Serum allergen-specific immunoglobulin E in cats with inflammatory bronchial disease. The study is novel as serum is largely not tested but intradermal testing is often used for evaluation of allergy. 

Overall, the article is well written and conclusions are well justified including authors explanation of caveats and limitations.

My only concern is under methodology authors have used Anti-Dog IgE but have not justified cross reactivity or lack thereof to full extent impacting ELISA outcomes. Please elaborate under discussion and cite relevant ref.

Author Response

Thank you very much for this feedback. According to your concern we have inserted the following paragraph:

Line 337-342: Another point of criticism is the fact that recombinant anti-canine IgE was used for the IgE measurement. This condition is because for cats the folding and thus the function of the recombinant proteins may differ from native proteins. Previous study also used anti-canine IgE, as feline IgE has not been available [15]. Other studies use human FcEpsilon receptors to detect IgE [13,14]. In the future, it could be investigated whether IgE measurements with recombinant feline IgE achieve different values.